# Fine-tuning protein language models boosts predictions across diverse tasks

Robert Schmirler [1,2,3] ✉, Michael Heinzinger [1] & Burkhard Rost [1,4,5]

Prediction methods inputting embeddings from protein language models have reached or even surpassed state-of-the-art performance on many protein prediction tasks. In natural language processing fine-tuning large language models has become the de facto standard. In contrast, most protein language model-based protein predictions do not back-propagate to the language model. Here, we compare the fine-tuning of three state-of-the-art models (ESM2, ProtT5, Ankh) on eight different tasks. Two results stand out. Firstly, task-specific supervised fine-tuning almost always improves downstream predictions. Secondly, parameter-efficient fine-tuning can reach similar improvements consuming substantially fewer resources at up to 4.5-fold acceleration of training over fine-tuning full models. Our results suggest to always try fine-tuning, in particular for problems with small datasets, such as for fitness landscape predictions of a single protein. For ease of adaptability, we provide easy-to-use notebooks to fine-tune all models used during this work for per-protein (pooling) and per-residue prediction tasks.

How to speak protein? Transformer-based[1] language models (LMs) have revolutionized Natural Language Processing (NLP). Large language models (LLMs) now perform at or above average human level. The newest generative models (GPT4[2] or PaLM2[3]) expand upon NLP through impressive capabilities in coding, math, and even common sense reasoning[4]. The success of LLMs has led to their widespread application from computer vision[5] over time series forecasting[6] to biologic language models[7–13]. For instance, protein language models (pLMs) are trained on many protein sequences[14–17]. PLMs learn from large data sets without any experimental annotation other than the sequence. The information extracted by the pLM, more precisely the value describing the last hidden layers, dubbed the embeddings, can be readily transferred to any protein-related prediction task. This generality makes pLMs suitable to a wide variety of prediction tasks spanning from secondary structure[14] over membrane regions[18], intrinsic disorder[19], protein structure[16,20], and protein-protein interaction[21] to predictions of stability[22,23] or solubility[24]. Successful applications to more function-related predictions include the identification of paratopes[25], epitopes[26], and signal peptides[27], as well as,

other tasks, e.g., related to the effect of sequence variation[28–31]. Embedding-based predictions seem particularly advantageous when experimental data are very limited[32]. Effectively, the embeddings from pLMs condense the understanding of the language of life[7,14]. Over the last 30 years, the de facto standard in protein prediction has been the use of evolutionary information, i.e., information from Multiple Sequence Alignments (MSAs) as input to machine learning[33]. Now, pLM-based predictions have reached and often even superseded the MSA-based state-of-the-art (SOTA) expert devices for many prediction tasks. Embeddings can be input to artificial feed-forward (ANN) or convolutional neural networks (CNN). More complex architectures have also been explored[34]. Continued unsupervised training can focus models on specific protein families[35] or enrich embeddings with structural information essentially creating a bi-lingual pLM[36]. Training specialist models from scratch on smaller, specific proteins, e.g., antibodies[25,37], seems an alternative to continued training (train pLM on large generic data and refine on specific proteins).

In contrast to the training recipes described above, which extract static representations from the pLM's last hidden layer without

[1]TUM (Technical University of Munich), School of Computation, Information and Technology (CIT), Faculty of Informatics, Chair of Bioinformatics & Computational Biology - i12, Garching/Munich, Germany. [2]TUM Graduate School, Center of Doctoral Studies in Informatics and its Applications (CeDoSIA), Garching/Munich, Germany. [3]AbbVie Deutschland GmbH & Co. KG, Innovation Center, BTS IR LU, Ludwigshafen, Germany. [4]Institute for Advanced Study (TUM-IAS), Garching/Munich, Germany. [5]TUM School of Life Sciences Weihenstephan (WZW), Freising, Germany. ✉e-mail: robert.schmirler@tum.de

changing its weights, it was shown for NLP that finetuning the parameters of the LLM is beneficial for downstream prediction performance[38–40]. This aligns well with the fact that the last layer's representation is not always optimal[41,42] and finetuning gives access to information stored in any layer of the model. However, for pLMs, finetuning remains less studied than for NLP, although some prediction tasks have been reported to profit from supervised, task-specific fine-tuning[21,43–45]. Here, we evaluated the impact of task-specific supervised fine-tuning by adding a simple ANN as a prediction head on top of the pLM encoder and applied supervised training to both, the pLM encoder and the prediction head. We compared the results to predictions using static, pre-trained embeddings (no finetuning of the pLM). For the larger models (ProtT5[14], ProstT5[36], both Ankh[15] models, and ESM2 3B[16]), we utilized Low Rank Adaptation (LoRA)[46], a particular version of a more general approach known as PEFT (Parameter Efficient Fine-Tuning)[38]. Freezing most of the model and updating only a small fraction of the weights (Table 1) accelerates training and prevents catastrophic forgetting[47,48]. This will become especially relevant, as pLMs[17] follow the NLP trend where bigger models usually translate to better downstream performance[17]. We assessed diverse prediction tasks from eight previously established benchmarks through the lens of three pLMs from which we derived some rules of thumb (and scripts) that simplify navigating the vastly growing space of PEFT methods applied to pLMs.

## Results and Discussion
### Fine-tuning is mostly successful
We trained 615 individual prediction methods (295 for fine-tuning, 320 using frozen embeddings from pre-trained pLMs - protein Language Models) comprising eight models (Table 1), each trained on eight different data sets (Table 2). We trained each model-task combination multiple times with different random seeds and all results constituted averages over those runs. The corresponding validation set selected the best training stop. For each prediction task (PT), we compared the performance between fine-tuning and pre-training (1). For ProtT5-XL-U50[14] (labeled ProtT5) and all five tested ESM2 versions[16] (differing in parameter size between 8 M (8*10^6) and 3B (3*10^9)), not all improvements were statistically significant within the 95% confidence interval (CI: Methods). Nevertheless, supervised fine-tuning numerically increased performance for almost all combinations (Fig. 1, detailed results in supplementary online material (SOM) Tables S1–S6). The exceptions were ESM2-150M applied to Stability prediction, and both Ankh[15] models. Ankh gained significantly by fine-tuning only for the mutational landscape data (GFP, AAV, and GB1: blue in Fig. 1).

For these data, performance relied less on transfer from model pretraining (Fig. S7) and mainly depended on the underlying transformer architecture. This might explain why Ankh performed similarly to ProtT5 and the ESM2. For the diverse data sets, this was not the case.

Two major factors differentiate Ankh from the other pLMs. Firstly, the T5[49] masked span pre-training differs from that of BERT-like[50] objective used for the other models. Secondly, the training procedure and architecture of Ankh was optimized using data (GFP, GB1, subcellular location, and secondary structure) also utilized in this work[15]. This might have reduced the ability to fine-tune these models.

For five of the 64 pLM/task combinations (tiles in Fig. 1), fine-tuning performed worse. The observation ESM2-150M on stability (Fig. 1 red tile) originated from instability in training picking a sub-optimal model (Fig. S5). The other four originated from the Ankh pLM family on disorder and secondary structure. We were not able to track down a root cause here but suspect that the different nature of the pre-training plays a role.

### LoRA was competitive with alternative PEFT methods
For ProtT5 and sub-cellular location prediction, we compared three parameter-efficient fine-tuning methods to LoRA[46]. Not having sufficient resources to do this analysis for all prediction tasks/pLMs, we chose this problem due to its limit in size and because of the success of fine-tuning on this problem (configuration in Method and Fig. 2). The fraction of trained model parameters were 0.25% for LoRA, 0.28% for DoRA[51], 0.12% for IA3[52] and 0.5% for Prefix tuning[53]. Despite these differences, runtimes for training and testing (inference) were within ±10% between methods, except for DoRA which was about 30% slower than the other three. In terms of prediction performance, LoRA and DoRA outperformed IA3 and Prefix-tuning (Fig. 2). Overall, all fine-tuning methods improved, on average, over pre-trained embeddings (61.3% from Table S5). As no method improved significantly over the well-established LoRA, we used it throughout our experiments. Of course, these results for a single model and dataset must not hold true in general. We encourage to explore parameter efficient fine-tuning of pLMs, utilizing new combinations of high-quality datasets, state-of-the-art models, and PEFT methods in future work and hope the notebooks made available by us help to pursue this research more easily.

### Insignificant gain for secondary structure prediction
For per-residue, three-class secondary structure prediction (helix, strand, other), fine-tuning improved only slightly (Fig. 2a; up to 1.2 percentage points for CASP12[54] and NEW364[14]). We confirmed this for the general-purpose ProtT5[14] and the bilingual, structure-tuned ProstT5[36]. Two effects might have hindered substantial improvement. Firstly, secondary structure might already have been captured in unsupervised pre-training. In fact, embeddings already capture some aspects of inter-residue contact formation[10,20]. Secondly, performance may have reached an upper limit[55]. One limitation of the benchmark is highlighted by the two data sets (CASP12[54] and NEW364[14]). Both were introduced to gauge the performance for unknown proteins. Other than that CASP12 is much smaller (12 proteins vs. 364) implying higher

## Table 1 | Protein language models (pLMs) applied in the study*

| Model | Architecture (pretraining) | Number of parameters (encoder) | Trained parameters LoRA | Encoder layers | Emb size | Huggingface model checkpoint |
|---|---|---|---|---|---|---|
| Ankh Base | Encoder-Decoder | 736 M | 2100 K | 48 | 768 | ankh-base |
| Ankh Large | | 1900 M | 4900 K | 48 | 1536 | ankh-large |
| ProtT5 | | 1200 M | 3500 K | 24 | 1024 | prot_t5_xl_uniref50 |
| ProstT5 | | 1200 M | 3500 K | 24 | 1024 | ProstT5 |
| ESM2 8 M | Encoder | 8 M | 163 K | 6 | 320 | esm2_t6_8M_UR50D |
| ESM2 35 M | | 35 M | 483 K | 12 | 480 | esm2_t12_35M_UR50D |
| ESM2 150 M | | 150 M | 1600 K | 30 | 640 | esm2_t30_150M_UR50D |
| ESM2 650 M | | 650 M | 3500 K | 33 | 1280 | esm2_t33_650M_UR50D |
| ESM2 3B | | 3000 M | 7700 K | 36 | 2560 | esm2_t36_3B_UR50D |

*Emb size provides the dimension of the embeddings of the corresponding pLM. Throughout the paper, we used the standard acronyms K for 10^3, M for 10^6, and G for 10^9.

**Table 2 | Task-specific datasets***

| Prediction level | Sequence diversity | Prediction task/data set | Number of sequences | | | Average length (number of residues/ tokens) |
|---|---|---|---|---|---|---|
| | | | Train | Validation | Test | |
| per-protein | Mutational landscapes | GFP[22,66] | 21,446 | 5362 | 27,217 | 237.0 |
| | | AAV[23,67] | 28,626 | 3181 | 50,776 | 736.3 |
| | | GB1[23,68] | 2691 | 299 | 5743 | 265.0 |
| | Diverse datasets | Stability[22,69] | 53,614 | 2512 | 12,851 | 45.0 |
| | | Meltome[23,70] | 22,335 | 2482 | 3134 | 544.5 |
| | | SubCellLoc[34,71] | 9503 | 1678 | 490 | 519.9 |
| per-residue | | Disorder[19,56] | 1056 | 118 | 117 | 118.1 |
| | | SecStr[14,73] | 9712 | 1080 | 364 | 255.0 |

*Prediction level: Per-protein predictions make a single prediction for an entire protein; per-residue predictions provide one number for each residue (position) in a protein. Sequence diversity: distinguishes between tasks with experimental data specific for individual proteins (mutational landscapes) and those for which data mixes different proteins from different organisms. Prediction task: as described in Methods; SubCellLoc: sub-cellular location, SecStr: secondary structure prediction (in 3 states: helix, strand, other). A number of sequences (i.e., proteins, note this is NOT the number of samples, e.g., for secondary structure prediction N proteins – number given in table - correspond to over 200*N residues in the data set): typical cross-validation using Train to optimize fine-tuning, Validation to optimize hyperparameters, and Test only to assess performance (Methods).

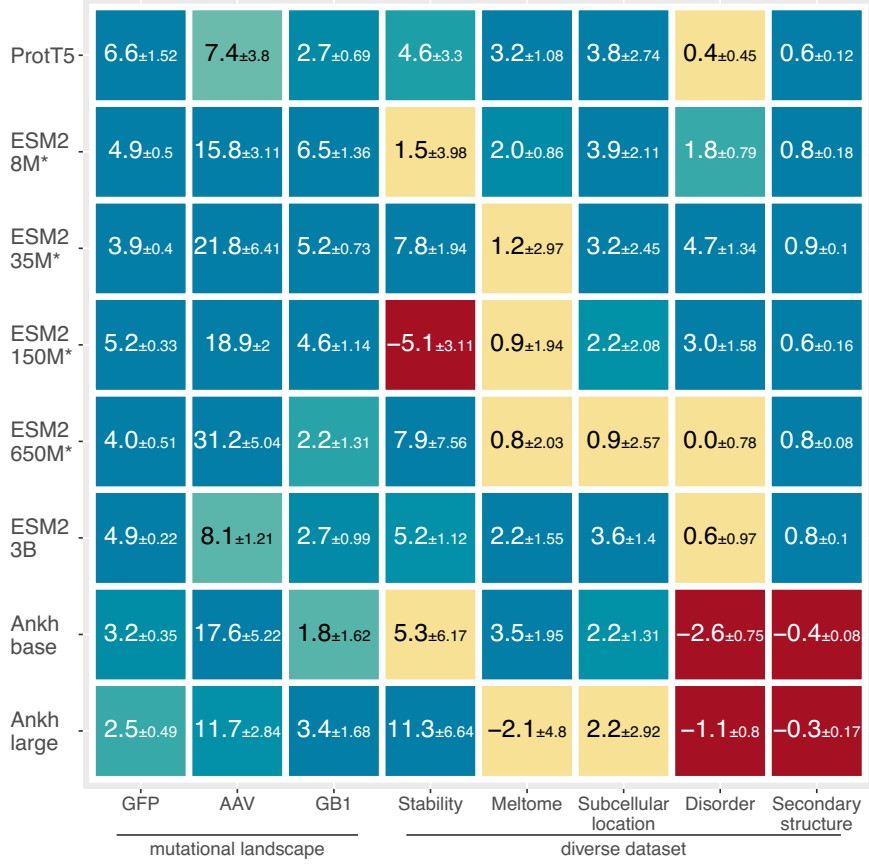

**Fig. 1 | Fine-tuning improved for most pLMs and tasks.** Asterisks (*) mark fully fine-tuned models; the others were LoRA-optimized (SOM Fig. S1). Values reflect percentage differences between the fine-tuned and pre-trained models (1) for the eight prediction tasks (x-axis). We had to use different performance measures, namely the Spearman rank correlation (GFP, AAV, GB1, stability, meltome and disorder), 10-class accuracy (Q10: sub-cellular location), and 3-class per-residue accuracy (Q3: secondary structure). Each tile compares fine-tuning to raw embeddings for one task. Blue tiles mark statistically significant increases (>1.96 standard errors; fine-tuning better), yellow tiles mark statistically insignificant changes (0 lies within the error margins of ±1.96 stderr) and for red tiles supervised fine-tuning significantly decreased performance. Error estimates (±percentage values) represent the 95% confidence intervals (CI, Methods). Source data are provided as a Source Data file.

statistical errors, there seems no a priori reason for choosing one over the other, and no method compared here is expected to have any systematic bias toward any of the two. Thus, the difference between both should estimate the statistical error. In other words, boot-strapping error estimates should be similar to the difference between the two sets. This was not the case at all (Fig. 3a: differences between CASP12 and NEW364 exceeded standard errors marked by distributions). Arguably, secondary structure prediction assessment is the best-solved task in protein structure prediction since decades[33,55]. Even for this relatively trivial problem, such a simple dichotomy seems not easily resolvable. In fact, the standard mantra: larger data sets without redundancy appears not to solve this dichotomy. These results

underscore how difficult it is to just plug in standard data sets to assess the performance of prediction methods without updating data and adapting it to advancing methods.

## Fine-tuning boosted disorder prediction

PLM-based SETH[19] reached the level of MSA-based SOTA methods, such as ODiNPred[56] in the prediction of per-residue protein disorder as described by CheZOD scores[56]. SETH inputs ProtT5 embeddings into a two-layer CNN.

Keeping those hyper-parameters and adding LoRA fine-tuning (dubbed SETH-LoRA), improved performance by 2.2 percentage points (from Spearman 0.72 to 0.736, Fig. 3b). Fine-tuning the much smaller 150 M parameter ESM2 model (Spearman: 0.742) improved overall solutions compared (Fig. 3b), including its larger counterparts (ESM2 with 650M/3B parameters, Table S4). Compared to SETH-LoRA,

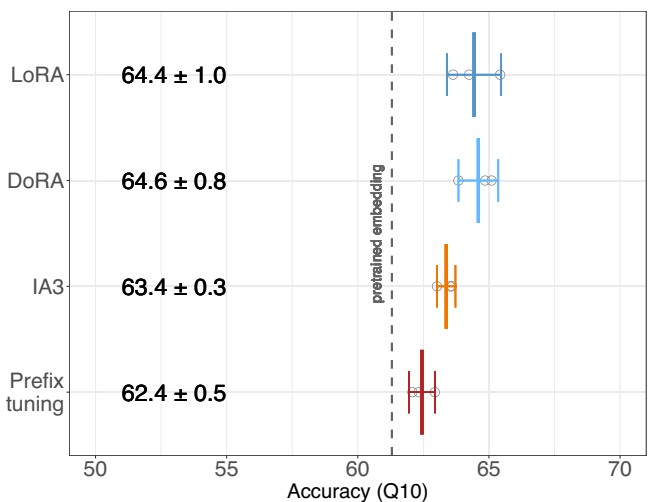

**Fig. 2 | Comparison of different PEFT methods.** ProtT5 model assessed on the prediction of sub-cellular location (x-axis: 10-state per-protein accuracy Q10). Mean values as well as 95% confidence intervals are computed from three training re-runs for each of the four PEFT methods: LoRA[46], DoRA[51], IA3[52], and Prefix-tuning[53]. We used the same configuration for LoRA and DoRA. The IA3 target modules were the key, value, and feed-forward layers. Prefix-tuning used 20 virtual tokens with 1024 dimensions to fit the ProtT5 dimensions. Circles represent individual training results. Differences between methods are mostly insignificant, with all four numerically outperforming the pre-trained embedding predictor on average (dashed grey line). Source data are provided as a Source Data file.

where only 2.5 million out of its 1.2 billion parameters are trained, for ESM2-150M all parameters were fine-tuned. Both approaches (2.5 m for ProtT5 vs 150 m for ESM2) performed similarly (Fig. 3b).

## LoRA topped pooling for subcellular location

Most predictions of subcellular location input signals averaged over entire proteins (e.g., amino acid composition). Embedding-based solutions do this through pooling, i.e., through embeddings derived from averaging over all intrinsic residue-level embeddings[14]. Light Attention (LA) substantially improves over such coarse-grained averaging by learning the optimal per-residue signal and combining this with the average[34]. LoRA fine-tuning combined the advantage of a small model (fewer free parameters) with the learned, weighted averaging of LA. Thereby, LoRA fine-tuning numerically surpassed LA, although the difference was statistically significant only at an 88% confidence interval (CI and not at the more common CI = 95% Table S9).

## Fine-tuning better-captured effects of mutations

For predicting mutation landscapes (Fig. 1 leftmost three columns) fine-tuning any pLM succeeded substantially. As differences between fine-tuned models were small (Fig. S3), we averaged performance across all fine-tuned pLMs (Fig. 4, for individual values refer to Table S2), and compared to homology-based inference (HBI, using MMseqs2[57] search) and to reference-free analysis (RFA[58]). RFA fits a decent first-order model for the fitness landscape reflecting some mutations for GB1 (protein G domain B1[59]; all possible variants for four residues, i.e., at four positions). For AAV2[60] (adeno-associated virus 2) for which a much larger 28-residue window was mutated, RFA performed less well. For GFP (green fluorescent protein[61]) the RFA analyses failed because some specific substitutions XnY (amino acid X at position n mutated to Y) occurred only in the test set. The fact that smaller and larger models performed alike on these tasks raised the prospect of using small, fine-tuned pLMs as computationally affordable, high-quality solutions for protein engineering.

## LoRA was substantially faster for larger models

The main drivers for the amount of computational resources required for model training were the parameter sizes of pLMs along with quadratic scaling of the attention mechanism (more resources for longer proteins). More recent GPUs used for LLM training (anything beyond 40GB of memory) will have sufficient memory to allow usage of all pLMs tested here. For less powerful hardware (Fig. 5b), mixed precision training nearly halved the required GPU memory without performance loss (both Ankh models were exceptional, as they do not support mixed precision training). Where GPU memory still was a

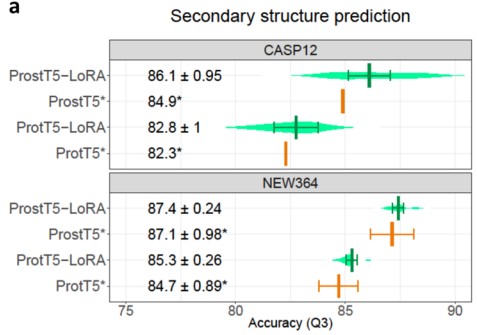

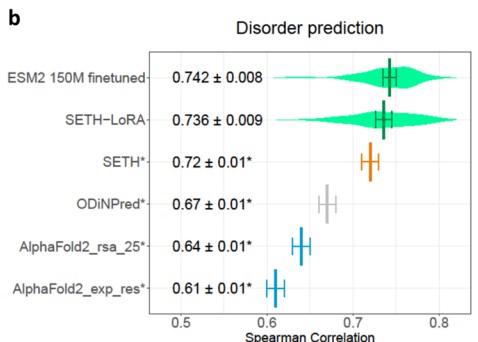

**Fig. 3 | Disorder prediction better, secondary structure prediction not.** Mean values and 95% confidence intervals (CI) were estimated through bootstrapping ($n = 10$ for a, $n = 25$ for b), violin plots reflect the data distribution. Source data are provided as a Source Data file. **a** Values for the pre-trained models (ProtT5[14] and ProstT5[36]) taken from literature[36] (no CI available for CASP12) and marked by asterisk (*); fine-tuning in green, pre-trained embeddings in orange. We included

two previously used data sets (CASP12[54] and NEW364[14]) to highlight the limitation of benchmarks. **b** Intrinsically disordered residues can be proxied by CheZOD scores[56]. The x-axis shows the Spearman correlation between experimental and predicted CheZOD scores for six methods. Values marked by asterisks (*) taken from the literature[19]. Fine-tuning results in green, pLM-based without MSA (SETH[19]) in orange, MSA-based SOTA in gray[56,72], and MSA-based AlphaFold2[76] in blue.

bottleneck, we applied gradient accumulation to reduce the actual on-device batch size as far as needed. When even an on-device batch size of 1 was insufficient, we used DeepSpeed to offload the optimizer and potential parameters to CPU-reduced GPU memory requirements further. As a trade-off, both gradient accumulation and CPU offloading slowed down training. Hence, both should be used cautiously. Implementing all these measures, we could fine-tune most pLMs tested here even on older GPUs with as little as 8GB memory (Fig. 5b). Unintuitively, both full model fine-tuning and parameter-efficient LoRA fine-tuning required the same amount of GPU memory and only differed in training speed (Fig. 5a) when CPU offloading was utilized. Embedding creation required much less GPU memory rendering it feasible even for datasets with very long sequences (Fig. S1).

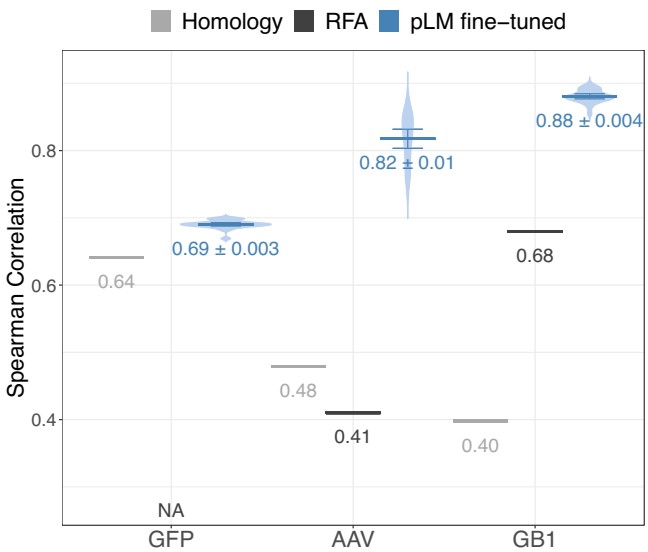

**Fig. 4 | Simple methods limited for mutational effects.** Blue: average performance across all fine-tuned pLMs (mean values with 95% CI, *n* = 24) with violin plots providing the underlying distribution; Gray: two simple baseline methods: Homology (HBI): MMseqs2[35] inferred the fitness value of test set proteins from most sequence-similar homolog in the training set. RFA (reference-free analysis[56]) fits models based on the assumption, that most of the mutational effects can be described as the sum of low-order effects. Source data are provided as a Source Data file.

## Fine-tuning recipe

To ease the simplicity of fine-tuning pLMs for your data set, we added the following recommendations. Before starting model training, dataset splits to measure model generalization and prevent over-estimating performance[23,28] are essential. First off: you need at least three data sets: training (optimizing weights), cross-training/validation (optimization of hyper-parameters, e.g., to decide between CNN and ANN), and testing (only touched to estimate performance). Typically, all entities in the test set (e.g. proteins) should adhere to the same split required between training/validation and testing. In particular, proteins have to be made non-redundant. This requires clustering by sequence identity using standard alignment methods such as MMseqs2[57] (simpler solutions tend to lead more likely to information leakage). For structure-related tasks, redundancy is best removed through 3D clustering as realized by Foldseek[62]. To optimize the prediction of mutational landscapes for a single protein, it might be best to train on k-mers with k = 1 (single amino acid variants) and test on k-mers with k > 1[22,23] (although this approach might focus more on avoiding over-fitting than on generating the best optimal model).

To predict landscapes of mutational effects for specific proteins, a challenge encountered in protein engineering, we recommend to first fine-tune a smaller pLM (pre-trained embeddings were limited: Fig. S3). Optimize hyperparameters and head architectures on this smaller model. If done, you could explore additional improvements from larger pLMs. For the fine-tuning on diverse tasks, larger mostly out-performed smaller models (Figs. S3 & S4). Therefore, starting with raw embedding-based solutions to identify the best model and to then investigate different prediction heads appeared better than optimizing the fine-tuning directly. Applying parameter-efficient LoRA fine-tuning and optimizing hyperparameters for the selected model afterward, will probably lead to an even better solution.

For our tasks, over-fitting mostly originated from data set characteristics. On the one hand, given a data set prone to over-fitting (e.g. too small, uninformative, or complex), neither hyperparameter nor model optimization could fully avoid the trap. On the other hand, for data sets not prone to over-fitting the training of fine-tuning was stable regardless of other factors. These factors affected raw embedding-based and fine-tuned models alike. Avoiding imbalanced datasets, providing sufficient high-quality training data, and choosing smaller models for limited data sets could mitigate over-fitting (SOM Sections 9 and 10).

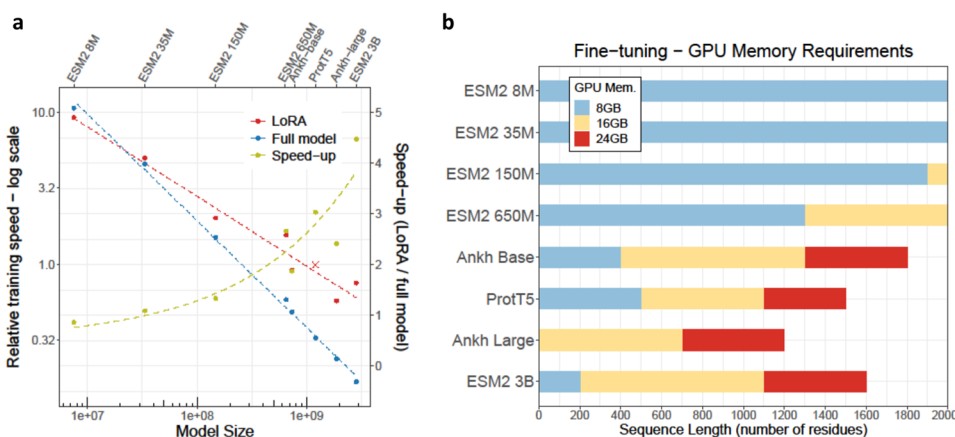

**Fig. 5 | Fine-tuning training speed and GPU requirements. a** Relative training speed of full fine-tuning (blue) and LoRA (red) is shown on a logarithmic scale, ProtT5 LoRA fine-tuning served as reference speed with value of 1 (x). The resulting speed-up for each model (olive) is shown on a normal scale. Experiments were performed with arbitrary sequences of length 1024 in a per-protein setting. For the smallest model (ESM2 8 M), LoRA fine-tuning was marginally slower than training the entire model. The larger the model, the more advantageous LoRA became. For the largest model (ESM2 3B), LoRA was about 4.5-fold faster. Panel **b** shows the maximum sequence length before the GPU runs out of memory (for 8, 18, and 24GB GPUs). All values obtained for memory-efficient training (mixed precision training, gradient accumulation with on-device batch size 1, and DeepSpeed CPU off-loading). Experiments were done for per-protein predictions, but memory requirements for per-residue training will be similar. Results valid for full-model and LoRA fine-tuning. Source data are provided as a Source Data file.

On the computational side, we recommend mixed precision training. Gradient accumulation and DeepSpeed's CPU-offloading should only be reserved to avoid memory constraints. With all these measures in place, a single 16 GB GPU enables fine-tuning in many cases (Fig. 5). Comparing different common PEFT methods (Fig. 2) did not suggest a clear winner. The established LoRA[46] method was among the best solutions and was stable across our experiments. The codebase provided by us simplifies experimenting with different PEFT approaches as it utilizes the Hugging Face PEFT[63] framework. We encourage you to compare different PEFT methods for your specific use cases. PEFT is memory efficient, but CPU-offloading could achieve the same. However, PEFT is also most compute-efficient for larger pLMs (Fig. 5a); it stabilizes training (SOM Section 5) and it renders saving model checkpoints orders of magnitude more memory efficient, as only the trained parameters need to be stored. Thus, we recommend LoRA fine-tuning for all models larger than ESM2 150 M (Fig. S2). We see little reason not to fully fine-tune smaller models.

In our hands, different random initialization seeds significantly altered results. These random variations reached the magnitude effect of hyperparameters or even model selection.

## Concluding thoughts
We applied fine-tuning[38,46] to a diversity of prediction tasks clearly showing improvements, on average. The extent of this improvement varied by task and pLM/model and was impacted by the amount of training data (Fig. S10), dataset balance (Table S8), models size (Fig. S3), and initial representation quality (Fig. S4).

Overall, our results revealed the gains initially observed in NLP from supervised task-specific fine-tuning of LLMs[39,40] to also apply to large protein LMs (pLMs). Supervised fine-tuning unlocks additional degrees of freedom in the predictor models. The last hidden layer has been optimized for the unsupervised pre-training objective (learning to reproduce masked sequences). This optimization might be suboptimal for any downstream task[41,42]. PEFT (or finetuning in general) enables information from middle layers to flow to the last layer, making it accessible to downstream tasks. Additionally, for per-protein predictions, the LoRA optimization may have learned weighted pooling of the last hidden layer, and that improved significantly over average pooling[34]. Lastly, the transformer models might extract additional information directly from the task-specific training. Randomly initialized smaller ESM2 models supported this view (Fig. S7, Table S17).

Therefore, we suggest to add supervised fine-tuning whenever applying transfer-learning, i.e., when inputting pLM embeddings into subsequent supervised prediction tasks. Our results suggested that you will most often benefit from this. To ease this additional step, we provided all our resources and added step-by-step recommendations.

## Methods
### Pre-trained pLMs
The pLMs used differed in size and architecture (Table 1), ranging from 8 million (ESM2 8 M) to 3 billion parameters (ESM2 3B). ESM2 pLMs are RoBERTa[64] based encoder models trained using an unsupervised masked language modeling objective[16]. The other three pLMs are built on T5[49], an encoder-decoder LLM pre-trained applying span masking[14,15]. We initialized our models using the pre-trained checkpoints available on Huggingface.

### Data
The data sets differentiated two aspects: prediction task level and sequence diversity (Table 2). The prediction task level collected cases of per-residue (e.g. secondary structure) and per-protein (e.g. sub-cellular location) prediction. Sequence diversity is distinguished between data sets with many sequence-diverse proteins and those from mutational studies analyzing single proteins through deep mutational scanning (DMS also known as MAVE) experiments[65].

Mutational landscapes: these data described fitness landscapes for three individual proteins: the green fluorescent protein (GFP[61]), the adeno-associated virus 2 capsid protein VP-1 (AAV2[60]), and the first binding domain of the protein G (GB1[59]). All three constitute regression tasks that measure prediction performance by ranking the correlation between the predicted and the experimentally measured property for each set. For GFP the property/fitness was measured through fluorescence intensity (experimental data[66], data split[22]). Training and validation set sequences were all within Hamming distance 3 (i.e. all variants up to three changes from wild type). For the AAV task, fitness was measured as the viability for packaging DNA payloads by mutating 28-amino acid window[67]. We used the 2-vs-rest data split from the FLIP benchmark[23] (variants with ≤2 in training and validation sets, those with more in the test set). The GB1 fitness score measures stability and binding affinity. The original experiment[68] mutated four positions, while we took the three-vs-rest data split from FLIP for easier comparability to existing benchmarks.

Per-protein prediction tasks included three prediction tasks. The first two sets focused on stability prediction formulated as regression tasks in analogy to the fitness landscapes. Stability predictions were assessed on measurements of protease susceptibility to digestion by de novo-designed mini-proteins[69]. We reused the TAPE data split[22] in which training and validation sets contain sequences from four design cycles while the test set holds neighborhoods at 1-Hamming distance to 17 promising candidates. Meltome utilizes data from measuring thermostability for proteins from 13 species[70]. We used the mixed split from the FLIP benchmark[23], which clusters proteins at >20% pairwise sequence identity (PIDE) through MMseqs2[57]. Excluding any pair within the same cluster from test and train/validation set, is the minimal means to reducing potential information leakage. The third data set included DeepLoc[71] data (incl. training/validation/testing splits) along with a novel test data set (setHARD[34]). The task is to predict the sub-cellular location in one of ten classes. MMseqs2[57] removed all sequence pairs at >20 PIDE between training, validation, and test sets.

Per-residue prediction tasks included disorder and secondary structure. Disorder predictions used CheZOD data[56] from nuclear magnetic resonance (NMR) spectroscopy. We used previously published data splits[19] clustering at <20 PIDE with MMseqs2[57]. The task is to predict CheZOD scores[72] which quantify the level of intrinsic disorder for each residue in a protein (each amino acid position) through a continuous scale. We bench-marked secondary structure predictions through data sets provided by NetSurfP-2.0[73] distinguishing three classes (H: helix, E: strand, and C: other/non-regular). In splitting these data, we used a recent split with more stringent redundancy-reduction, along with another test set (NEW364)[14].

Performance measures were copied from the data set developers (Table 2). All regression tasks were evaluated using Spearman rank correlations, i.e., all three mutational landscapes (GFP, AAV, GB1), both stability-related data sets (Stability and Meltome), as well as, the per-residue regression of Disorder. For the classification tasks, accuracy was defined as a 10-class per-protein accuracy (Q10) for sub-cellular location, and as a 3-class per-residue accuracy (Q3) for secondary structure (for a detailed per-class analysis see SOM Section 2).

### Model training
Top-level comparisons contrasted pre-trained to fine-tuned models as follows. For the pre-trained results, we generated embeddings for all data sets. For per-protein tasks, we averaged over the sequence length; for each protein, this yielded a vector of dimension 1 x embedding_size. For per-residue tasks, we used all residue embeddings and their labels; this resulted in vectors of the same dimension as for the per-protein task (1 x embedding_size), albeit this time for each residue (i.e., L*embedding_size for a protein of length L). Next, we trained a single fully connected layer of size 32, inputting exclusively these embeddings and outputting either a single value (regression) or going into

another output layer with one neuron for each possible output class, followed by a softmax layer to get a probability distribution. Reaching a plateau in training loss terminated training. We repeated each training five times with different random seeds. For fine-tuning, we added the same fully connected layer (size 32) to the pLM encoder as a prediction head. For ProtT5[14] and Ankh[15], we used average pooling of the last hidden states over the sequence length dimension during training on per-protein tasks. Following the author's advice for ESM2[16], we connected the prediction head only to the very first token (special token <CLS>). The training was repeated three times with different random seeds. Training terminated when training and validation loss flattened out. To increase training efficiency and save time, we applied Parameter Efficient Fine-Tuning (PEFT)[38] to all models. In particular, we applied the Low-Rank Adaptation (LoRA)[46] implementation of PEFT. For the smaller ESM2 models (up to the 650M version), we could investigate full model fine-tuning (SOM Section 3 and Fig. S1).

For embedding and fine-tuning we reported the performance of the checkpoint with the lowest validation loss for each run. Finally, we simply computed the percentage differences between the fine-tuned and pre-trained models (1):

$$\Delta(\text{PT}) = \text{performance(PT)}_{\text{finetuned}} - \text{performance(PT)}_{\text{pretrained}} \quad (1)$$

We also performed a limited hyperparameter optimization at the beginning (Table S11). Once selected, the hyperparameters were frozen for most comparisons (Tables S10 and S12). We used the Adam optimizer[74] with default parameters. For LoRA fine-tuning we reused a previously suggested configuration[52], namely rank 4, alpha 1, applied to query, key, value, and the output of the attention layers. A minimal LoRA rank of four has also been suggested previously[21] for pLMs. To realize the batch size (Table S12) for fine-tuning, we applied gradient accumulation as needed given our hardware. Initially, we fine-tuned models on full precision but switched to mixed precision for larger models. Embedding generation used half precision. The Ankh models only support full precision for both. All training ran on a single NVIDIA A10G GPU with 24GB. We used Torch version 1.13.1 with transformers version 4.26.1.

Training times for fine-tuning depended crucially on the available GPU. Nevertheless, the following basic trends were notable. Training times were mostly driven by model and dataset size, as well as, the quadratic scaling with sequence length. As all these factors accumulate, smaller models and shorter sequences led to much faster training (Fig. 5a). For instance, for the small Disorder data (average sequence length: 118 residues), fine-tuning the full ESM2-8M took 12 minutes, while the much larger ProtT5 (LoRA fine-tuning) took 190 minutes (16-fold increase). On the other end, for the AAV data (average length: 736 residues with nearly 30k proteins), ESM2-8M training took 130 minutes while a single ProtT5 training ran over 4K minutes (>30-fold increase). The time to train an embedding-based prediction method was mostly determined by the time needed to compute embeddings because the relatively small predictor models that we used required negligible runtime. The creation of embeddings took approximately as long as fine-tuning the same model for a single epoch (typically we needed 5-50 epochs, Table S12).

Per-residue secondary structure: We fine-tuned five models for ProtT5[14] and another five for ProstT5[36], initializing with different random seeds. For ProstT5, we added the prefix token <AA2fold> to each sequence to code the input type (amino acid rather than structure as ProstT5 is a bilingual pLM). We trained for five epochs and calculated the validation loss at the end. For both models (ProtT5 and ProstT5), utilizing the same two-layer CNN as applied previously[14,36] for pretrained embeddings to simplify comparisons. Of the five, we selected the model with the lowest validation loss and measured performance on common data sets (CASP12[54] and NEW364[14]). Bootstrapping established confidence intervals (Fig. 2a).

Per-residue disorder fine-tuning stacked up two model variants to compare to other methods[19]. SETH-LoRA used ProtT5 with the same two-layer CNN as the original SETH. ESM2 150M reused the ESM2 setup from the top-level evaluation (last hidden states of <CLS> token with single dense layer prediction head). For both variants, we trained five models with different random seeds for ten epochs, and calculated the validation loss twice per epoch, selecting the model with the lowest validation loss out of 100 checkpoints (5 random seeds, 10 epochs, 2 points per epoch). Bootstrapping provided confidence intervals (Fig. 3b).

Per-protein subcellular location: For the 10-class classification, we reused the single-layer dense network from our top-level evaluation. We trained five models with different random seeds for five epochs, calculating validation loss twice per epoch. We selected the model with the lowest validation loss in each run, and then calculated Q10 accuracy and standard errors from all five models on setHARD[34] and reported the averages (Table S9).

Fitness landscapes contrasted fine-tuned pLMs to two baselines, namely, homology-based inference (HBI) and reference-free analysis (RFA)[58]. We averaged test performance overall fine-tuned pLMs (checkpoints with lowest loss on validation set) for the three mutational landscape data sets (GFP, AAV, GB1).

HBI: MMSeqs2[57] searched each query (Q) in the test set against all proteins training set proteins. The fitness value of the top training hit (closest homolog) predicted that for Q.

RFA: We applied the R implementation of RFA (version 1.0.0) without modification to the GB1 data. For AAV, we removed sequences not containing 735 residues (the algorithm failed on insertions and deletions). This reduced the training data by about 1%. RFA failed for GFP because some substitutions in the test set were missing from the training set. For AAV and GB1, we fitted a first- and second-order model and reported results for the better of the two.

## Statistics & Reproducibility

Replication of computational results is made possible by defining random states (seed). These random seeds are given in the detailed results (Table S1 and S2). The code and data provided by us will allow complete reproduction of our results, in addition, results for each individual model training run are provided with the source data.

During our research, we excluded a single training run (Finetuning ESM2 150M on the AAV dataset, seed 98) which was a clear outlier, for details check Table S2.

Since random data splits lead to a large overestimation[23] of predictor performance, we used dataset splits from previous work (Table 2) specifically designed to test for generalization.

## Reporting summary

Further information on research design is available in the Nature Portfolio Reporting Summary linked to this article.

# Data availability

All data sets analyzed are freely available through the original sources:
- GFP and stability: https://github.com/songlab-cal/tape
- AAV, GB1, meltome and secondary structure: https://github.com/J-SNACKKB/FLIP
- Sub-cellular location: https://github.com/HannesStark/protein-localization
- Disorder: https://github.com/DagmarIlz/SETH

Easing access, we re-packaged all data at https://github.com/RSchmirler/data-repo_plm-finetune-eval[75]. When using those data, please quote and consult the authors of the original data sets. All data generated for this study and source data to generate figures and tables is also available in this repository. Source data are provided with this paper.

# Code availability

We made the notebooks for fine-tuning all our models (Table 1) for per-protein and per-residue tasks available at[75]. By default, these

notebooks use LoRA fine-tuning; full-model fine-tuning is optional. To create prediction methods based on pre-trained pLM embeddings, we provided two additional notebooks (one to generate embeddings, and another to train predictors) with sample data in the same repository. RFA v1.0.0 R scripts are available from https://github.com/whatdoidohaha/RFA.

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

## Acknowledgements

Thanks to Nikita Kugut (TUM) for support with many aspects. M.H. and B.R. were supported by the Bavarian Ministry of Education through funding to the TUM, by a grant from the Alexander von Humboldt Foundation through the German Ministry for Research and Education (BMBF: Bundesministerium für Bildung und Forschung), and by a grant from Deutsche Forschungsgemeinschaft (DFG-GZ: RO1320/4-1). All computational resources for this work were provided by AbbVie. Last, but not least, thanks to all those who maintain public databases and make their resources publicly available.

## Author contributions

R.S., M.H. and B.R. contributed to the conception of this study. R.S. performed all experimental work, results analysis, and creation of figures. The first draft of the manuscript was written by R.S. R.S., M.H. and B.R. commented on and refined subsequent versions of the manuscript.

## Funding

## Competing interests

R.S. is an employee of AbbVie. The design, study conduct, and financial support for this research were provided by AbbVie. AbbVie participated in the interpretation of data, review, and approval of the publication. M.H. and B.R. declare no competing interests.
