## [Peer Review File · Nature Communications]

Fine-tuning protein language models boosts predictions across diverse tasksReviewer #1 (Remarks to the Author):

Main contribution:

Similar to the impressive capability of large language models (LLMs) in natural language processing (NLP), protein language models (pLMs) have seen several successes in protein-related prediction tasks and become the "foundation models" for state-of-the-art solutions. Nevertheless, PLMs are unsupervised models and, in many applications, need to be adapted or fine-tuned to problem-specific regression or classification tasks. However, directly fine-tuning the entire pLM is computationally expensive.

This work studies 8 tasks where 3 pLMs (ESM2, ProtT5, Ankh) are applied and shows two main conclusions: (a) fine-tuning is beneficial as compared to using a pLM as a pre-trained feature encoder (b) PEFT could reach similar results with fewer resources.

This line of research does carry substantial significance.

I have two main concerns about the manuscript in its current form and which require a major revision of the paper:

(a) The PEFT part is very under-explored and lacks crucial information. Especially, there is very little about "resources", so we cannot learn exactly what "fewer" means.

(b) The paper is purely empirical, so rather than just show a bunch of tables with results, if they could prescribe a recipe for when to fine-tune, best practices for fine-tuning or using PEFT, that are actionable for readers, that would be a useful contribution.

(c) It is lacking some details to reproduce the results. Maybe the github repositories contain these in the code, but I am mentioning some below.

Point-wise comments (not in the order of importance):

1. Can the authors explain why some of the other prominent structure prediction models like AlphaFold, RoseTTAFold2 were not studied in these settings?

2. The authors do limited hyper-parameter optimization, but with deep learning models, these can make a significant performance difference. The supplementary section 2 on hyper-parameter tuning does not discuss what ranges of values were tried. It only mentions which one they chose. Did they see any overfitting: i.e. validation loss starts increasing, as this is a frequent problem with finetuned models.

3. There is no mention of the LoRA parameter optimization, like the LoRA's hyperparameter alpha. See usage in proteomic tasks in [1].

4. There is no mention of what supervised head architectures were used for the pretrained models. Did they try simple linear models too, like logistic regression by using the embeddings as features? Was it a multi-layer perceptron?

5. There does not seem to be a discussion of which weight matrices from LoRA were adapted and whether they tried various combinations like (key, value), (query, key) etc.

6. While the original LoRA paper does not consider FFN matrices, people do adapt those matrices [2] to reach better performance.

7. It will be helpful if they can add text, on overall, on how much of an impact the improvement from finetuning can give, in practice? I understand that the tasks are very diverse, but the sense that metrics like Precision and Recall give, where you see the trade-off, is hard to get from Spearman correlation from 0.72 to 0.736 for instance. It seems a bit unfair to issue a blanket statement saying "fine-tuning is better" unless it is a significant margin.

8. There needs to be a section on compute time: how long does it take for the fine-tuned models vs the pretrained models so that practitioners can get a trade-off between these and make decisions based on their needs. Add info on training time and inference time per example, maybe?

9. In Table 3, can they add info on: while fine-tuning (all layers) for these models, and for inference, how much GPU memory would be needed. In the text, add info on what are the memory sizes of typical GPUs used by academic labs (e.g., V100, 32G; A100, 40/80G; A6000, 48G).

10. It would help to show a plot on the tradeoff between GPU memory and prediction performance (x-axis is % of parameters fine-tuned, y-axis is the performance).

11. Which version of PyTorch/Huggingface and what datatype (FP32 or FP16) is used for the experiments? This is important to consider when understanding efficiency.

12. It would also help to look at this other work as it is very related and cite it: [3]

13. Since this is a Nat-Comm submission, not a submission to a ML conference, and given that this work is highly empirical: no novel methods, no novel strategies in fine-tuning, or recipes to get a better model beyond the vanilla options (which also haven't been explored and hence this is a valuable study), it feels lacking. Here are options:

(a) what is it in fine-tuning that is helping get better models. Is it just more parameters being tuned? Is it the architecture that is helping? Is there a sense of how many layers to fine-tune before you see *some* improvement in performance?

(b) a biological case study of what this work impacts. Pick one or two of the tasks and show examples where benefits were seen and in what aspect.

[1] Bo Chen et al., xTrimoPGLM: Unified 100B-Scale Pre-trained Transformer for Deciphering the Language of Protein, 2023

[2] Edward J Hu et al., LoRA: Low-Rank Adaptation of Large Language Models}, ICLR, 2022

[3] <https://www.ncbi.nlm.nih.gov/pmc/articles/PMC10659351/>

Minor:

Typo in Figure 2(a)

Reviewer #2 (Remarks to the Author):

Summary of the work: The paper "Fine-tuning protein language models boosts predictions across diverse tasks" by Schmirler and colleagues presents a comprehensive study on the impact of fine-tuning protein Language Models (pLMs) on various protein prediction tasks. The authors explore the benefits of task-specific supervised fine-tuning across three state-of-the-art pLMs (ESM2, ProtT5, Ankh) over eight different prediction tasks. The study highlights two primary findings: task-specific fine-tuning generally enhances prediction performance, and parameter-efficient fine-tuning achieves comparable improvements with significantly fewer resources. The research underscores the utility of fine-tuning pLMs for protein prediction tasks.

Strengths:

This work is highly timely. Many pLMs have been recently published for various tasks and it is unclear whether fine-tuning approaches can improve the performance of these models.

This a highly rigorous benchmark, employing 8 different tasks and 7 variations of 3 state-of-the-art pLMs (Ankh Base, Ankh Large, ProtT5, ESM2 8M, ESM2 35M, ESM2 150M, ESM2 650M) to assess the impact of fine-tuning on prediction quality.

Two key findings are highlighted: task-specific supervised fine-tuning generally improves predictions, and parameter-efficient fine-tuning achieves similar enhancements with fewer resources.

It explicitly investigates the use of Low Rank Adaptation (LoRA), a popular parameter-efficient fine-tuning approach very popular for NLP applications, but relatively unknown in the pLM world. LoRA only updates a minimal fraction of the model's weights, significantly reducing the computational requirements and reducing the likelihood of catastrophic forgetting.

Overall, the paper supports the broader application of fine-tuning in bioinformatics, underlining the general applicability and efficiency of fine-tuning in protein prediction models.

Software is mostly available, although see later comment.

Weaknesses:

The authors only explore the use of LoRA for the T5-based models (ProtT5, ProstT5, and Ankh models), while all weights are fine-tuned for ESM2 models. Could the authors discuss and justify this choice?

The effectiveness of fine-tuning varies by task, pLM, and model, indicating that although fine-tuning generally enhances performance, significant disparities remain unexplained. For example, for the largest tested ESM2 model (ESM2 650M), 3 out of 8 tasks showed negligible performance gains, while about half of the tasks tested on both Ankh-based models resulted in negligible or negative performance improvements. This variation suggests a need for additional analysis to determine the specific conditions under which fine-tuning yields statistically significant benefits.

To investigate why fine-tuning works better for certain tasks and models but not for others, the authors could consider several approaches:

1. Task characteristics and input data: For those tasks where fine-tuning shows significant improvement, are there specific patterns in task difficulty (structure, function, interaction) or data characteristics (sequence length, diversity, motifs) that might explain the variance in fine-tuning benefits?
2. Model architecture and parameters: Can the architecture and number of parameters (e.g. model size, number of layers, type of attention, or other task-specific adaptations, etc) explain the observed differences in fine-tuning performance gain? Similarly, how does the choice of hyperparameters such as learning rate, batch size, and the amount of fine-tuning data affect the gain in performance?
3. Representation quality: Could fine-tuning's varied effectiveness relate to pre and post-fine-tuning representation quality? Models with high-quality initial representations might show minimal fine-tuning benefits. Testing this hypothesis could involve assessing feature quality through simple models focusing on key features before and after fine-tuning, or examining performance declines after synthetically removing or corrupting important features. Alternatively, selective ablation, targeting samples with particular features, could further provide insight into feature significance before and after fine-tuning.
4. Data leakage: The paper mentions potential data leakage between training and testing, particularly in the context of the Ankh models, however, little discussion is shared in the supplementary file. This information is crucial and should be further discussed in the main text. Can data leakage also explain the insignificant gains in performance in other pLMs and models?
5. Related to the last one, the risk of overfitting is only briefly discussed in the supplementary file, but it is an important point that deserves a detailed discussion in the main paper including strategies to detect and prevent overfitting in the context of fine-tuned models.

I am not suggesting that the authors perform all these analyses, but I encourage them to investigate a few of them in the hope of gaining a deeper understanding of when fine-tuning will bring a performance gain and when it will not.

Computational complexity: The document does not explicitly discuss the computational complexity or memory requirements of fine-tuning large pLMs. While a rigorous study might be outside the

scope of this paper, some quantitative information would be useful for the reader to decide on a fine-tuning strategy.

Open-source code: The authors have made available a GitHub repository containing notebooks for ProtT5 fine-tuning (<https://github.com/agemagician/ProtTrans/tree/master/Fine-Tuning>). They mention a Colab library that can be adjusted to fine-tune ESM2 models. Similarly, they mention a GitHub library that can be adapted to fine-tune Ankh or ProtT5 with LoRA. To improve reproducibility, it would be beneficial for the authors to share the complete pipeline and training parameters for all tasks and models. This also includes the prediction heads used for the different tasks and models.

Reviewer #3 (Remarks to the Author):

Summary.

This paper is dedicated to investigating the downstream fine-tuning performance of diverse pre-trained protein language models. The authors study three protein language models (ESM2, ProT5, Ankh) on eight downstream tasks. They consider both full fine-tuning and parameter-efficient fine-tuning (PEFT) in their experiments. The results tell that (1) task-specific supervised fine-tuning almost always improved downstream predictions; (2) PEFT could reach similar performance improvements by consuming substantially fewer resources.

Pros.

1. The authors conduct extensive experimental studies and consider eight downstream prediction tasks and three pre-trained protein language models.
2. Multiple runs are performed and errorbar estimation is provided.
3. Sufficient details, codes, and configurations are provided for reproducibility.

Cons.

1. It is well known that the full fine-tuning performance highly relies on appropriate tuning configurations and the number of tuning samples. Detailed studies of tuning configurations (such as learning rate, tuning iterations, batch size, weight decay, etc.) and the amount of tuning data are missing.
2. As indicated in Figure 1, the fine-tuning significantly hurts model prediction performance in five (red) settings. Any reasons behind it?
3. As for parameter-efficient fine-tuning techniques, soft prompt tuning, adapter, prefix, and Lora are the most representative ones. It is needed to investigate the other three methods, for a more convincing and consistent conclusion.

Response to Reviewer #1

Q (a) The PEFT part is very under-explored and lacks crucial information. Especially, there is very little about "resources", so we cannot learn exactly what "fewer" means.

A Sorry for this oversight. We have added the number of trainable LoRA parameter for each model to Table 1. We now provide details about the LoRA configuration in Methods.

The comparison of the plethora of available PEFT methods exceeds the scope of this work focused on assessing the effect of fine-tuning and showing the applicability of PEFT to pLMs and computational biology. However, we have added a comparison of different PEFT methods to our Results (Fig. 2) to confirm LoRA being a good choice.

Overall, pLMs have also shown to be not particularly sensitive to hyperparameter changes, neither in our work not that of others (S Sledzieski et al & JL Ferres bioRxiv doi [10.1101/2023.11.09.566187](https://doi.org/10.1101/2023.11.09.566187)). Nevertheless, we agree that identifying optimal PEFT strategies would be beneficial. However, as the variation between models, data sets, task types and even between training runs is considerable, it may be difficult to draw generalized implications. We encourage future research in this direction and feel our code provides a good starting point for such experiments.

Q (b) The paper is purely empirical, so rather than just show a bunch of tables with results, if they could prescribe a recipe for when to fine-tune, best practices for fine-tuning or using PEFT, that are actionable for readers, that would be a useful contribution.

A Thank you for this idea, we agree this is a useful addition, especially for lowering the barrier for future users, potentially, even expanding towards users with less ML but stronger biology background. We added an actionable recipe to our Discussion with best practices to train pLM based predictors. Along with the sample code that we now provide, this will hopefully help others to apply PEFT to their own work.

Q (c) It is lacking some details to reproduce the results. Maybe the github repositories contain these in the code, but I am mentioning some below.

A Thanks & sorry. All data and code is now in the repository https://github.com/RSchmirler/data-repo_plm-finetune-eval/tree/main. We also added previously missing information in the Methods section (more details at the respective comments below).

Q Can the authors explain why some of the other prominent structure prediction models like AlphaFold, RoseTTAFold2 were not studied in these settings?

A Thanks for this question! In our work, we explored how far we can *tweak* foundation models (pLMs) into specific directions by training on specific prediction tasks. While the structure prediction models are very powerful tools, they are not foundation models, but already geared towards a very specific direction (predicting structure). Fine-tuning those models is therefore mostly interesting in the context of domain adaptation to specific protein families.

Neither AlphaFold2 (nor the just published AlphaFold3 for that matter), nor RoseTTAFold2 are based on pLMs. The answer gets a little more murky for ESMfold which you didn't explicitly list in your question but which aspires to competing with both the methods you mentioned by using pLMs as input rather than MSAs. However, ESMfold and other related

solutions tap into many of the concepts published by DeepMind that actually achieve performance increase through pLM-independent modules, i.e., for those our LoRa story still wouldn't provide relevant lessons. To the best of our knowledge, this leaves us only with the in-house tools EMBER2 and EMBER3D that predict 2D/3D structure directly from pLM embeddings. These we didn't include because we have stopped working on those tools simply because they are not competitive, yet. Put simply, at this point, they can only compete in terms of prediction speed (10^6 times faster than AF2), but remain too inaccurate to ascertain that LoRA lessons would hold once those tools have been raised to a competitive level.

More generally, while AF2/3, ColabFold, OpenFold, and RoseTTAFold2 are so powerful structure prediction tools that they are changing protein biology completely, pLMs continue to address rather different objectives. For instance, embedding-based tools complement AF2 et al in predicting the membrane region in the predicted structure, and the effect of sequence variation. More generically, pLMs provide protein-specific predictions where AF2 et al optimize family-averaged predictions.

Q The authors do limited hyper-parameter optimization, but with deep learning models, these can make a significant performance difference. The supplementary section 2 on hyper-parameter tuning does not discuss what ranges of values were tried. It only mentions which one they chose. Did they see any overfitting: i.e. validation loss starts increasing, as this is a frequent problem with finetuned models.

A Thanks for pushing us substantially further! We have added the information about the hyperparameter search (SOM Table S11) and explored several additional factors impacting training (incl. the new figures 2 and several display items in SOM). We do not claim that the parameter used by us will be optimal for all cases but they worked well enough during all our experiments to suggest them as a default starting point. As already mentioned in the first question, we found pLM fine-tuning rather insensitive to hyperparameters (S Sledzieski et al & JL Ferres bioRxiv doi [10.1101/2023.11.09.566187](https://doi.org/10.1101/2023.11.09.566187)). In fact, the data/task combination itself (redundancy, data set splits, representativeness, generalizability, difference between prediction and experimental accuracy) seems to play a much more important role in influencing performance and training behavior. We saw some overfitting for smaller and especially imbalanced datasets. We have added more detailed investigations of overfitting in SOM 8 and SOM 9.

Q There is no mention of the LoRA parameter optimization, like the LoRA's hyperparameter alpha. See usage in proteomic tasks in Bo Chen et al (2023) xTrimopGLM.

A As explained above (Q(a)): we could not systematically optimize the LoRA parameters. However, we HAVE added some experiments toward this end (e.g. Fig. 2).

Q There is no mention of what supervised head architectures were used for the pretrained models. Did they try simple linear models too, like logistic regression by using the embeddings as features? Was it a multi-layer perceptron?

A Sorry for the lack of clarity. The architecture is now mentioned in Methods 4.3.

Next, we trained a single fully connected layer of size 32, inputting exclusively these embeddings and outputting either a single value (regression) or going into another output layer with one neuron for each possible output class, followed by a softmax layer to get a probability distribution. Reaching a plateau in training loss terminated training. We repeated each training step five times with different random seeds.

For fine-tuning, we added the same fully connected layer (size 32) to the pLM encoder as a prediction head.

We took this small multi-layer perceptron to be simple enough and did not consider adding linear models. We found repeatedly in previous work that linear models were doing worse (e.g. Ilzhöfer, D. et al & Rost, B. doi [fbinf.2022.1019597](https://doi.org/10.1101/2022.10.19.597)). We therefore mirror existing prediction methods for embedding-based predictors to give them a fair shot when comparing to fine-tuning.

Q There does not seem to be a discussion of which weight matrices from LoRA were adapted and whether they tried various combinations like (key, value), (query, key) etc.

A Thanks, we provide this information now in Methods. We applied adapters to key, query, value and output within the attention layers, and did not investigate other configurations, this setting was also suggested in previous work (<https://doi.org/10.48550/arXiv.2205.05638>). Others investigated this specifically for pLMs and found that at least the value matrix W_v should be adapted for optimal results (Table 3 [<https://doi.org/10.1101/2023.11.09.566187>]).

Q While the original LoRA paper does not consider FFN matrices, people do adapt those matrices (EJ Hu et al (2022) LoRA... ICLR) to reach better performance

A Thanks for mentioning this. We provide this information now in *Methods* along with all the other LoRA parameters. As mentioned above, we adapted not only weight matrices but also the output FFN layer within the attention blocks. We focused our work on the variation between models and datasets and therefore left LoRA parameters unchanged during our experiments.

Q It will be helpful if they can add text, on overall, on how much of an impact the improvement from finetuning can give, in practice? I understand that the tasks are very diverse, but the sense that metrics like Precision and Recall give, where you see the trade-off, is hard to get from Spearman correlation from 0.72 to 0.736 for instance. It seems a bit unfair to issue a blanket statement saying "fine-tuning is better" unless it is a significant margin.

A Thank you, point very well taken! We agree and added a more detailed investigation for the classification tasks (SOM Section 2). This showed balanced gains across prediction classes. For the regression tasks we feel that Spearman correlation is already the most informative measure. Other correlation coefficients are not beneficial (as ranking is the most interesting here) and error metrics like MSE are not informative for arbitrary fitness scores.

Q There needs to be a section on compute time: how long does it take for the fine-tuned models vs the pretrained models so that practitioners can get a trade-off between these and make decisions based on their needs. Add info on training time and inference time per example, maybe?

A True, thank you! We agree and included a section on training times in *Methods*. We also added a measure of relative training speed between models in Fig. 5.

Q In Table 3, can they add info on: while fine-tuning (all layers) for these models, and for inference, how much GPU memory would be needed. In the text, add info on what are the memory sizes of typical GPUs used by academic labs (e.g., V100, 32G; A100, 40/80G; A6000, 48G).

A Thanks, we added a resource section to Results and provide GPU memory requirements for finetuning (Fig. 5b) as well as embedding creation (Fig. S1), which is also what you need for

inference.

Q It would help to show a plot on the tradeoff between GPU memory and prediction performance (x-axis is % of parameters fine-tuned, y-axis is the performance).

A Thanks for this suggestion. From our results we have no reason to believe that such a trade-off exists. While there certainly is a correlation between model size and performance (shown in SOM Section 7), we show in SOM Section 5, that there is no systematic difference in performance between finetuning full models and LoRA finetuning (we use rank of 4, which is at the lower end in respect to tuned parameters), so we also do not expect to see any systematic effect between those two extremes. We suggest using LoRA to make training faster and more stable for large models but not to directly increase performance over full model finetuning.

Q Which version of PyTorch/Huggingface and what datatype (FP32 or FP16) is used for the experiments? This is important to consider when understanding efficiency.

A Thanks! We added the information to methods. We used Torch version 1.13.1 with transformers version 4.26.1. We found that mixed precision training does not lose performance vs. full precision. So, we recommend this in general. A disadvantage of the Ankh models is, that they do not support this.

Q It would also help to look at this other work as it is very related and cite it: [3].

A Thank! We completely agree and now refer to their work in several places.

Q Since this is a Nat-Comm submission, not a submission to a ML conference, and given that this work is highly empirical: no novel methods, no novel strategies in fine-tuning, or recipes to get a better model beyond the vanilla options (which also haven't been explored and hence this is a valuable study), it feels lacking. Here are options:

(a) what is it in fine-tuning that is helping get better models. Is it just more parameters being tuned? Is it the architecture that is helping? Is there a sense of how many layers to fine-tune before you see *some* improvement in performance?

(b) a biological case study of what this work impacts. Pick one or two of the tasks and show examples where benefits were seen and in what aspect.

A Thanks! We feel this study is valuable, as it shows the limits of pretrained pLMs (their embeddings) for downstream use, especially for mutational fitness prediction. We believe that providing practical instructions together with explanatory notebooks easily adaptable to new data, will prove valuable to the research community, especially for research groups less experienced in computational methods, i.e., a group of users highly enriched in those who will NOT attend ML conferences.

It has been shown (F-Z Li, AP Amini, KK Yang, AX Lu: *Pretrained protein language model transfer learning: is the final layer representation what we want?* MLSB) that the pretraining sequence reconstruction objective does not lead to a good representation for all tasks. Fine-tuning the attention allows the models to access this information from lower layers. Fine-tuning also lets models implicitly learn pooling per-residue embeddings for per-protein tasks. The latter was turned into a method specifically optimizing pLM-based prediction of subcellular location (H Stärk et al 2021, <https://doi.org/10.1093/bioadv/vbab035>).

In fact, this IS one of the reasons for picking this particular prediction task. LoRA performed on par with the highly specific method designed toward this end by simply fine-tuning!

The fitness landscape tasks chosen provide another example for a topic of high relevance for protein design/engineering both of which are immensely relevant to both academia and industry.

Overall, however, we chose NOT to choose just a few prediction tasks to exactly avoid cherry-picking. In contrast, we “cherry picked” a diversity of tasks hoping that exactly through this large diversity, our results become meaningful to a wider group of users. Ultimately, diversity bears generality. As it turns out, the devil was, as usual, in the detail: diversity taught us humbleness, i.e., that simple generic rules are difficult to come by. This message would have been lost if we had zoomed into only protein structure prediction, or only location, or only fitness landscapes.

Q Typo in Figure 2(a)

A Sorry, we couldn't find this typo. May be that originated from the figure at too low resolution, we tried to do better in R1.

Response to Reviewer #2

Q Task characteristics and input data: For those tasks where fine-tuning shows significant improvement, are there specific patterns in task difficulty (structure, function, interaction) or data characteristics (sequence length, diversity, motifs) that might explain the variance in fine-tuning benefits?

A Thanks for bringing this up. In short: we wished there were!

We found stark differences between mutational fitness landscapes and diverse datasets, with larger fine-tuning gains seen for the former (SOM Sections 7 & 11). On the one hand, small and imbalanced data sets size tended to perform poorly (SOM Section 10). For instance, the location prediction task is only hard for minority classes because so little data is provided for those (SOM Table S8). On the other hand, the over 3-order of magnitude smaller and more imbalanced location data performed better than that for secondary structure. The difficulty of predicting comes down to the combination of data and prediction task. With data available currently, conclusions about underlying biologic task difficulty are not well grounded. We have added several paragraphs to R1 which discuss these findings.

Q Model architecture and parameters: Can the architecture and number of parameters (e.g. model size, number of layers, type of attention, or other task-specific adaptations, etc) explain the observed differences in fine-tuning performance gain? Similarly, how does the choice of hyperparameters such as learning rate, batch size, and the amount of fine-tuning data affect the gain in performance?

A Model size explained a significant share of the differences (SOM Section 7). In terms of the other model specific parameters you mention, current SOTA models are very similar with only two underlying architectures (RoBERTa for ESM2 and T5 for the others). We find no categorical difference between those two, apart from the *Ankh* models responding less well to fine-tuning, possibly because the *Ankh* models have already been optimized using several prediction tasks.

One important conclusion – added to the *Conclusion* in R1 – is that fine-tuning seems surprisingly robust with respect to hyperparameter changes. The amount of data and pLM-model size are, on the other hand is very relevant (SOM Section 7 & 10).

Q Representation quality: Could fine-tuning's varied effectiveness relate to pre and post-fine-tuning representation quality? Models with high-quality initial representations might show minimal fine-tuning benefits. Testing this hypothesis could involve assessing feature quality through simple models focusing on key features before and after fine-tuning, or examining performance declines after synthetically removing or corrupting important features. Alternatively, selective ablation, targeting samples with particular features, could further provide insight into feature significance before and after fine-tuning.

A Yes, you are right on. Very much so! We added new results in SOM Section 8 and Fig. S4. In short, models with lower initial representations (worse performance from pretrained embeddings) are gaining more. As expected from our other results, this is very prevalent for mutational landscapes where all fine-tuned models reach the same performance. For the diverse tasks the trend is less clear and does not compensate for initially weaker performance.

Q Data leakage: The paper mentions potential data leakage between training and testing, particularly in the context of the *Ankh* models, however, little discussion is shared in the supplementary file. This

information is crucial and should be further discussed in the main text. Can data leakage also explain the insignificant gains in performance in other pLMs and models?

A Thanks for mentioning this. We rewrote this part to make it clearer. The alleged data leakage pertained to the optimization undertaken by the authors of Ankh: this is a foundation model type of pLM generated by unsupervised masking objective, but optimized by comparing downstream task performance. Some of the data sets (GFP, GB1, sub-cellular location, and secondary structure) used in this study had partaken in Ankh’s optimization. Comparing Ankh-large with ProtT5, which is the most similar model in size and architecture, using pretrained embedding results (SOM Table S5), Ankh is doing worse for *Stability*, *Meltome* and *Disorder* which it was not optimized for (only for AAV it does better than ProtT5 despite not having seen it before).

The bias introduced by this optimization might have led to the difficulties in fine-tuning *Ankh*, especially for diverse tasks. This is therefore not relevant for other models and does not explain insignificant gains for some tasks.

Q Related to the last one, the risk of overfitting is only briefly discussed in the supplementary file, but it is an important point that deserves a detailed discussion in the main paper including strategies to detect and prevent overfitting in the context of fine-tuned models.

A Thanks. We expanded the overfitting discussion in the SOM (adding SOM Section 10) and found that overfitting is mostly data related. Small and imbalanced data sets lead to overfitting. If sufficient high-quality data is available, no overfitting could be observed. We added this observation to our *fine-tuning recipe* section in *Results & Discussion*.

Q Computational complexity: The document does not explicitly discuss the computational complexity or memory requirements of fine-tuning large pLMs. While a rigorous study might be outside the scope of this paper, some quantitative information would be useful for the reader to decide on a fine-tuning strategy.

A Thanks for suggesting this. We added information about training times to Methods and a paragraph about (GPU) memory requirements and relative training speed in Results, including Fig. 5.

Q Open-source code: The authors have made available a GitHub repository containing notebooks for ProtT5 fine-tuning (<https://github.com/agemagician/ProtTrans/tree/master/Fine-Tuning>). They mention a Colab library that can be adjusted to fine-tune ESM2 models. Similarly, they mention a GitHub library that can be adapted to fine-tune Ankh or ProtT5 with LoRA. To improve reproducibility, it would be beneficial for the authors to share the complete pipeline and training parameters for all tasks and models. This also includes the prediction heads used for the different tasks and models.

A Thanks! We now make the entire training pipeline available through notebooks in the repository. Those support all models and training settings used in our work. Training parameters used for all experiments are given in the Methods and SOM Section 6. All datasets are also available in the repository.

Response to Reviewer #3

Q It is well known that the full fine-tuning performance highly relies on appropriate tuning configurations and the number of tuning samples. Detailed studies of tuning configurations (such as learning rate, tuning iterations, batch size, weight decay, etc.) and the amount of tuning data are missing.

A The number of tuning samples obviously has a strong impact, but more data (of high quality) is nearly always beneficial. We added these results (SOM Section 10). While deep learning methods often highly rely upon hyperparameter tuning, we found pLMs remarkably robust against a wide range of choices. For instance, our grid search for the ProtT5 model on the subcellular location data (SOM Table S11) revealed a stable performance, with 10 out of 12 hyperparameter sets still achieving a superior performance to the pretrained embedding prediction method on the same task (SOM Table S5). Only too low a learning rate in combination with too high batch sizes performed markedly worse. Similarly, pLMs have been found to be robust against LoRA parameter variation, recently (S Sledzieski et al & JL Ferres bioRxiv <https://doi.org/10.1101/2023.11.09.566187>).

Q As indicated in Figure 1, the fine-tuning significantly hurts model prediction performance in five (red) settings. Any reasons behind it?

A Just to identify those 5: ESM2-150M: Stability and Ankh-base, Ankh-large SecondaryStructure and Disorder (2x2=4). Overall, the *Stability* task did not show stable training behavior. Early stopping picks suboptimal model checkpoints for the ESM2 150M model by chance. This resulted in the red tile. In Fig. S5 fine-tuning is successful when looking at the ten best model checkpoints instead of applying early stopping.

The other four belong to the Ankh family, which does not seem to behave well when it comes to fine-tuning on diverse tasks. Optimizing Ankh embeddings for downstream tasks during pretraining, might have had negative effects on the ability to fine-tune these models. We added the following paragraph to R1:

For five of the 64 pLM/task combinations (tiles in Fig. 1), fine-tuning performed worse. The observation ESM2-150M on stability (Fig. 1 red tile) originated from instability in training picking a suboptimal model (Fig. S5). The other four originated from the Ankh pLM family on disorder and secondary structure. We were not able to track down a root cause here but suspect that the different nature of the pre-training plays a role.

Q As for parameter-efficient fine-tuning techniques, soft prompt tuning, adapter, prefix, and Lora are the most representative ones. It is needed to investigate the other three methods, for a more convincing and consistent conclusion

A We added a comparison of LoRA with prefix tuning, IA3 activation scaling and a recent LoRA variant DoRA to Results (Fig. 2). LoRA and its variant did best but overall differences were low. We also expanded the comparison between full model tuning and LoRA in SOM Section 5.

Reviewer #1 (Remarks to the Author):

The authors have addressed the concerns I had adequately. Re: citations, it seems that the following paper is now published at PNAS.

. Sledzieski, S. et al. Democratizing Protein Language Models with Parameter-Efficient Fine-Tuning

Reviewer #1 (Remarks on code availability):

I also checked the github repository and the notebooks for fine-tuning and generating embeddings look alright to me. I did not run any of the code however. I downloaded the dataset provided (training_data.zip) and it contains the train/test/val splits for the 7 tasks. I did not open the files, but I hope they contain the protein relevant info as well as the labels.

Reviewer #2 (Remarks to the Author):

I thank the authors for carefully and thoroughly addressing my comments and other reviewers' comments. One important question that is still not fully answered is under which conditions fine-tuning helps, and if so, how much improvement can be expected. While fully understanding this question will require follow-up investigations and is beyond the scope of this paper, the authors show some initial analysis indicating that the initial representation quality plays a significant role (Supp. Fig. 4). I thank the authors for this figure, which I find very interesting.

Overall, I recommend publication.

Minor comments:

- There are a few sentences that have strange constructions, e.g., "Fine-tuning mostly successful," "Initial representation quality important for diverse tasks" (the verb is missing in both sentences). I spotted other strange constructions. I recommend proofreading the manuscript with Grammarly or similar.
- The link to the GitHub repo in the abstract is broken (the link points to <https://github.com/RSchmirler/data->).

Reviewer #3 (Remarks to the Author):

Below is the second review based on the author's feedback.

1. The author has provided additional hyperparameter studies on tuning configuration and the amount of data. My concern is resolved.
2. The author has provided stability explanations about the inferior performance in the original Figure 1. The new results show that the fine-tuning is successful when looking at the ten best model checkpoints. I prefer this group of results.
3. Additional PEFT methods are studies. My concern is resolved.

Reviewer #3 (Remarks on code availability):

The codes are well-structured and have adequate details. I have not installed and run the codes.

Response to Reviewer #1

Q The authors have addressed the concerns I had adequately.

A Thank you again for your insights and suggestions, the additional focus on PEFT details has improved our study significantly.

Q Re: citations, it seems that the following paper is now published at PNAS. (Sledzieski, S. et al. Democratizing Protein Language Models with Parameter-Efficient Fine-Tuning.)

A Thanks for spotting this, we corrected this and cite the publication instead of the preprint now.

Q I also checked the github repository and the notebooks for fine-tuning and generating embeddings look alright to me. I did not run any of the code however. I downloaded the dataset provided (training_data.zip) and it contains the train/test/val splits for the 7 tasks. I did not open the files, but I hope they contain the protein relevant info as well as the labels.

A Thank you. Datasets indeed contain the needed sequences and labels

Response to Reviewer #2

- Q I thank the authors for carefully and thoroughly addressing my comments and other reviewers' comments. One important question that is still not fully answered is under which conditions fine-tuning helps, and if so, how much improvement can be expected. While fully understanding this question will require follow-up investigations and is beyond the scope of this paper, the authors show some initial analysis indicating that the initial representation quality plays a significant role (Supp. Fig. 4). I thank the authors for this figure, which I find very interesting.
- A Thank you again for suggesting this, we found this angle of investigation very helpful as well. We agree that this is promising avenue for future research.
- Q There are a few sentences that have strange constructions, e.g., "Fine-tuning mostly successful," "Initial representation quality important for diverse tasks" (the verb is missing in both sentences). I spotted other strange constructions. I recommend proofreading the manuscript with Grammarly or similar.
- A Thank you for the suggestion, we used Grammarly and modified the manuscript according to those suggestions.
- Q The link to the GitHub repo in the abstract is broken (the link points to <https://github.com/RSchmirler/data->).
- A Thank you for spotting this. We made sure to provide the complete link in our submission (to https://github.com/RSchmirler/data-repo_plm-finetune-eval).

Response to Reviewer #3

Q The author has provided additional hyperparameter studies on tuning configuration and the amount of data. My concern is resolved.

A Thank you for your comments, it has significantly benefited this work.

Q The author has provided stability explanations about the inferior performance in the original Figure 1. The new results show that the fine-tuning is successful when looking at the ten best model checkpoints. I prefer this group of results.

A We agree and prefer those results as well. To adhere to common practices and to point out the difficulties with early stopping, we leave the original Figure 1 in place

Q Additional PEFT methods are studies. My concern is resolved

A Thank you for bringing up this suggestion.

Q The codes are well-structured and have adequate details. I have not installed and run the codes.

A Thank you for checking the code as well.